# Quantum Dissipative Adaptation with Cascaded Photons

**Thiago Ganascini** †, **Thiago Werlang** † and **Daniel Valente** *,†

Instituto de Física, Universidade Federal de Mato Grosso, Cuiabá 78060-900 , Brazil;
ganascini@fisica.ufmt.br (T.G.); thiago.werlang80@gmail.com (T.W.)
* Correspondence: valente.daniel@gmail.com
† These authors contributed equally to this work.

**Abstract:** Classical dissipative adaptation is a hypothetical non-equilibrium thermodynamic principle of self-organization in driven matter, and it relates transition probabilities with the non-equilibrium work performed by an external drive on dissipative matter. Recently, the dissipative adaptation hypothesis was extended to a quantum regime with a theoretical model where only one single-photon pulse drives each atom of an ensemble. Here, we further generalize that quantum model by analytically showing that N cascaded single-photon pulses driving each atom still fulfill a quantum dissipative adaptation. Interestingly, we find that the level of self-organization achieved with two pulses can be matched with a single effective pulse only up to a threshold, above which the presence of more photons provides unparalleled degrees of self-organization.

**Keywords:** single-photon wavepacket; quantum thermodynamics; self-organization

## 1. Introduction

Finding non-equilibrium self-organization principles is a longstanding quest [1–3]. The various behaviors of different non-equilibrium dissipative systems may seem too diverse to obey one set of principles. Yet, as physicists, we have witnessed many times in history where unified theories of seemingly distinct phenomena have come to light. For instance, the Boltzmann distribution is a powerful principle relating the energy and the probability distribution for any system at thermal equilibrium with a weakly coupled reservoir. Inspired by that, as well as by living matter [3], the principle of dissipative adaptation as a non-equilibrium version of the Boltzmann distribution has recently been proposed and discussed [4–7].

According to the dissipative adaptation hypothesis, which was originally formulated with the help of stochastic thermodynamics [8], instead of looking at the probability of a given state (as dictated by the Boltzmann distribution), we can think in terms of the transition probabilities between pairs of states whenever driving forces arbitrarily far from equilibrium come into play. As a consequence of this change in perspective, both the energy of each individual state (as shown by the Boltzmann distribution) and the energy corresponding to the work consumed during the transition between a pair of states jointly affect the dynamics of the system. The key insight here is that trajectories of highest work consumption (absorption followed by dissipation) may tend to accumulate (the system receives enough energy to jump over a certain high barrier but dissipates this energy, becoming unable to return). That eventually allows the system to become finely tuned (adapted) according to the patterns of a given far-from-equilibrium drive. Mathematically, the transition probability $p_{i \to j}(t)$ between two generic states $i$ and $j$ fulfills [4]

$$\frac{p_{i \to j}(t)}{p_{i \to k}(t)} = e^{-\beta E_{kj}} \frac{p^*_{j \to i}(\tau - t)}{p^*_{k \to i}(\tau - t)} \frac{\langle e^{-\beta W_{\text{abs}}} \rangle_{ik}}{\langle e^{-\beta W_{\text{abs}}} \rangle_{ij}}, \tag{1}$$

where $k$ is another generic state. Here, $E_{kj} = E_j - E_k$ recovers the equilibrium Boltzmann distribution. $W_{\text{abs}}$ describes the (stochastic) work absorbed from the drive during the

dynamic history of the system (with the brackets indicating an ensemble average). The emphasis on the history, as conveyed by dissipative adaptation, is complementary to the conventional wisdom concerning the system's stationary states [1,9–11], and the work acquires a similar role to that of the system's energy in generating the tendencies of the system towards its many possible non-equilibrium configurations. Finally, $p^*$ stands for the time-reversed trajectory probability, which can be seen as a kind of kinetic (rather than thermodynamic) term accounting for the likelihood of back-and-forth jumps between a given pair of states. As far as self-organization is concerned, it is still an open debate whether kinetics are more or less significant than thermodynamics [12]. In summary, even if we consider the principle of dissipative adaptation to be a promising hypothesis for the emergence of life-like non-equilibrium self-organization in inanimate matter, with its emphasis on the relation between the history of work consumption and the transition probabilities, we understand that how it relates to other principles (such as kinetic asymmetry [12], maximum-entropy production [1,9–11], or even the high multistability analogous to equilibrium spin glasses [2]) and how broadly applicable it is are issues that are far from being settled.

To test whether the dissipative adaptation hypothesis is truly a general principle, we should also look for it at the most elementary scales of nature, where only a few energy levels are present, temperatures are very low, and quantum coherences and quantum fluctuations are significant. Indeed, a quantum dissipative adaptation was recently found in a case where the source of work was a single-photon pulse, for which quantum fluctuations were far more significant than the average field [13,14]. In its original formulation, quantum dissipative adaptation also depends on quantum coherence. A quite counterintuitive effect emerges in this case, namely, the highest average work performed by the photon does not correspond to the highest probability of single-photon absorption. Finally, the quantum version is valid at zero temperatures, a regime forbidden by its classical counterpart. However, both the classical and the quantum scenarios have shown evidence that work is a key thermodynamical quantity to "bias" the transition probabilities of a fluctuating system, hence determining the degree of stabilization (adaptation) to a given drive. A sketch of the quantum self-organization described here can be seen in Figure 1.

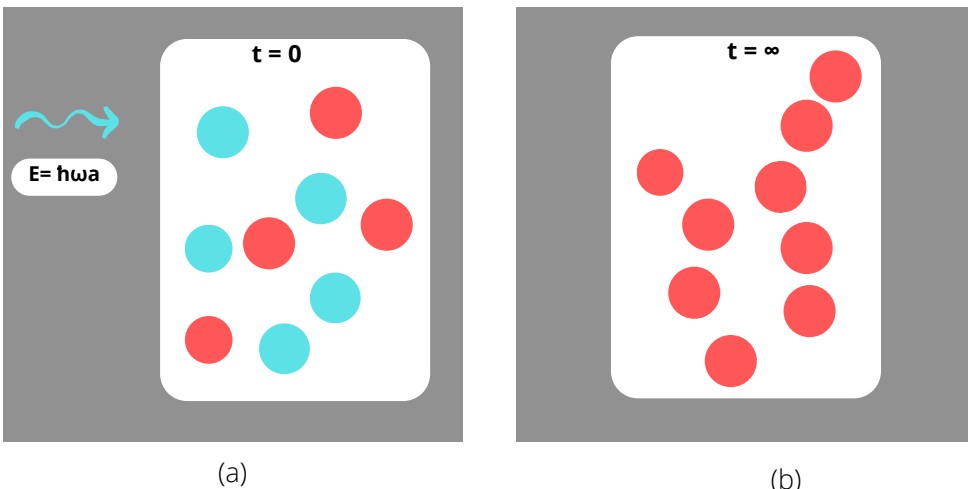

(a)  (b)

**Figure 1.** The quantum self-organization discussed here. (**a**) Initially, at time $t = 0$, an ensemble of $\Lambda$ atoms are in a mixed state, either in the ground state $|a\rangle$ (blue circles) or the ground state $|b\rangle$ (red circles), at temperature $T = 0$. Then, a non-equilibrium environment composed of single-photon pulses (at resonance $\omega_L = \omega_a$) starts driving the atoms non-perturbatively. (**b**) Eventually, after a long time $t \to \infty$, all of the atoms transition to the self-organized (pure) state $|b\rangle$ (red circles). Our goal in this paper is to understand the dynamics and thermodynamics of the process where $N$ photons drive each atom.

Here, we raise the question of how broadly applicable the idea of a quantum dissipative adaptation is. In particular, we are interested in the cases where quantum self-organization with a single photon is not ideal, i.e., when the transition probability between two ground states of a Λ atom is less than unity after the single-photon pulse has gone away. Is it true that the more energy is given (by means of more subsequent photon pulses arriving), the more the system adapts, so the higher the degree of self-organization? That would provide a significant analogy with living matter, where we also see an increase in specialization in time provided that there are enough resources. This adds to a flourishing list of other analogies between living matter and dissipative systems, such as self-replication [3,14,15] and the emergence of adaptive energy-seeking behavior [9,16], as well as photonic systems [17,18]. In this paper, we show that the relation between the transition probability and the average work as found for a single photon remains valid if $N$ consecutive pulses of arbitrary shapes reach every atom in the ensemble. Moreover, we show the conditions under which a "two-photon environment" may improve self-organization beyond any possible "single-photon environment". Our results represent a further step in the direction of promoting the dissipative adaptation hypothesis to a general thermodynamic principle of non-equilibrium self-organization. In particular, in our quantum model, kinetic asymmetry, which is described by the uneven spontaneous emission rates of the lambda atoms ($\Gamma_a \neq \Gamma_b$), is shown not to be the key to the success of self-organization in our system of interest, but rather thermodynamics (a sufficiently large number of polarized single-photon pulses in our case). It is also interesting to note that our results are not derived from Crooks' fluctuation theorem [8], contrary to the classical dissipative adaptation in Equation (1). Instead, since Crooks' theorem is not applicable at zero temperature, we employ the so-called system-plus-reservoir approach to open quantum systems. This suggests that the notion of dissipative adaptation (classical and quantum) is perhaps more fundamental and worthy of further investigation.

## 2. Materials and Methods

We consider a global time-independent Hamiltonian of the system and its environment,

$$H = H_S + H_I + H_E. \tag{2}$$

We model light–matter interaction via a dipole coupling in the rotating-wave approximation [13,19,20],

$$H_I = -i\hbar \sum_\omega (g_a a_\omega \sigma_a^\dagger + g_b b_\omega \sigma_b^\dagger - \text{H.c.}), \tag{3}$$

where $\sigma_k = |k\rangle\langle e|$ (for $k = a, b$), and H.c. is the Hermitian conjugate. Orthogonal polarization modes $\{a_\omega\}$ and $\{b_\omega\}$, respectively, interact with transitions $|a\rangle$ to $|e\rangle$ and $|b\rangle$ to $|e\rangle$. This is a multi-mode Jaynes–Cummings model, and our interest is in the limit of a broadband continuum of frequencies,

$$\sum_\omega \rightarrow \int d\omega \varrho_\omega \approx \varrho \int d\omega, \tag{4}$$

where we have defined a constant density of modes $\varrho_\omega \approx \varrho$. Physically, this means that there is no frequency filter in our idealized environment. The continuum of frequencies allows us to employ a Wigner–Weisskopf approximation to obtain the dissipation rates $\Gamma_a = 2\pi g_a^2 \varrho$ and $\Gamma_b = 2\pi g_b^2 \varrho$. The environmental Hamiltonian reads

$$H_E = \sum_\omega \hbar\omega (a_\omega^\dagger a_\omega + b_\omega^\dagger b_\omega). \tag{5}$$

The system is described by a three-level Hamiltonian with states named $|a\rangle$, $|b\rangle$, and $|e\rangle$ so that

$$H_S = \hbar\omega_a |e\rangle\langle e| + \hbar\delta_{ab} |b\rangle\langle b|. \tag{6}$$

Most importantly, when in $\Lambda$ configuration, these energy levels provide an elementary energy barrier (the excited state $|e\rangle$) separating two highly stable states (the two ground states $|a\rangle$ and $|b\rangle$). The energy difference between states $|b\rangle$ and $|a\rangle$, which is given by $\hbar\delta_{ab} = \hbar(\omega_a - \omega_b)$, although relevant for thermal equilibrium considerations, does not affect our results as far as non-equilibrium effects are concerned.

## 3. Results

### 3.1. Revisiting the Single-Photon Wavepacket Scenario

In Ref. [13], the initial state of the field was considered as a single-photon pulse added to a zero-temperature environment, namely,

$$|1_a\rangle = \sum_\omega \phi_\omega^a(0)\, a_\omega^\dagger |0\rangle, \tag{7}$$

where $|0\rangle = \prod_\omega |0_\omega^a\rangle \otimes |0_\omega^b\rangle$ is the vacuum state of all of the field modes. Note that the polarization is fixed as that of the $a_\omega$ modes, thus imposing an environmental condition to which the atom is expected to adapt. The level of adaptation will depend on the pulse shape, $\phi_\omega^a(0)$, which admits a spatially dependent representation

$$\phi_a(z,t) = \sum_\omega \phi_\omega^a(t) e^{ik_\omega z}. \tag{8}$$

For simplicity, our electromagnetic environment is one-dimensional in space, explaining why only coordinate $z$ appears. Also, the modes propagate towards the positive direction, with a dispersion relation in the form $k_\omega = \omega/c$. This idealized model closely approximates the experimental scenario known as waveguide quantum electrodynamics (waveguide-QED) [21,22]. If the photon is prepared by means of the spontaneous emission of a distant source atom (not shown in the model), then we would have that

$$\phi_a(z,0) = \mathcal{N}\Theta(-z)\exp[(\Delta/2 + i\omega_L)z/c], \tag{9}$$

where $\Delta$ is the linewidth of the pulse (the lifetime of the atomic source), and $\omega_L$ is the central frequency of the photon (the transition frequency of the atomic source). The size of the pulse in space is characterized by $c\Delta^{-1}$ in this case.

The key quantities analyzed in the context of dissipative adaptation are transition probabilities and their relations with the absorbed work from an external drive. In the quantum case, we are especially interested in the transition probability from $|a\rangle$ to $|b\rangle$,

$$p_{a\to b}(t) \equiv \langle b|\mathrm{tr}_E\Big[U|a,1_a\rangle\langle a,1_a|U^\dagger\Big]|b\rangle, \tag{10}$$

where $U = \exp(-iHt/\hbar)$, $H$ is the global Hamiltonian in Equation (2), and $\mathrm{tr}_E$ is the partial trace over the field modes. Taking advantage of the number conservation in the rotating-wave approximation, we restrict our model to the single-excitation subspace, which is parametrized by $U|a,1_a\rangle = \psi(t)|e,0\rangle + \sum_\omega \phi_\omega^a(t)a_\omega^\dagger|a,0\rangle + \phi_\omega^b(t)b_\omega^\dagger|b,0\rangle$. We now have that

$$\begin{aligned} p_{a\to b}(t) &= \sum_\omega |\phi_\omega^b(t)|^2 \\ &= (2\pi\varrho c)^{-1}\int_{-\infty}^\infty dz\,|\phi_b(z,t)|^2. \end{aligned} \tag{11}$$

To obtain the probability amplitude of the field, we make a Wigner–Weisskopf approximation, thus obtaining that

$$\begin{aligned} \phi_b(z,t) &= \phi_b(z-ct,0)e^{-i\delta_{ab}t} \\ &\quad + \sqrt{2\pi\varrho\Gamma_b}\,\Theta(z)\Theta(t-z/c)\psi(t-z/c)e^{-i\delta_{ab}z/c}, \end{aligned} \tag{12}$$

where $\Theta(z)$ is the Heaviside step function. This expression evidences the superposition of the freely propagating driving field with that emitted by the atom. Because

the incoming pulse is fixed to the $a_\omega$ modes, we have that $\phi_b(z - ct, 0) = 0$. By using Equations (12) and (11), we find that

$$p_{a \to b}(t) = \Gamma_b \int_0^t dt' |\psi(t')|^2,$$ (13)

after an appropriate change of variables. The meaning of Equation (13) is that, for the system to get to state $|b\rangle$ while departing from $|a\rangle$, the entire history of the excitation amplitude from time 0 to $t$ matters.

We can think of two options for maximizing that integration: (1) by increasing the excitation of the atom (making $|\psi(t)|$ as large as possible); (2) by increasing the time duration of the pulse (without necessarily exciting the atom too much; $|\psi(t)| \ll 1$).

To test options (1) and (2), we have to solve for the excitation amplitude. Employing the Wigner–Weisskopf approximation once again, we have that

$$\partial_t \psi(t) = -\left( \frac{\Gamma_a + \Gamma_b}{2} + i\omega_a \right) \psi(t) - g_a \phi_a(-ct, 0).$$ (14)

For $\psi(0) = 0$, this gives us

$$\psi(t) = -g_a \int_0^t \phi_a(-ct', 0) e^{-\left( \frac{\Gamma_a + \Gamma_b}{2} + i\omega_a \right)(t - t')} dt',$$ (15)

which provides a generic solution.

For the sake of definiteness, let us take the exponential profile from Equation (9), which gives

$$\psi(t) = -A e^{-\left( \frac{\Gamma_a + \Gamma_b}{2} + i\omega_a \right)t} \left[ e^{\left( \frac{\Gamma_a + \Gamma_b - \Delta}{2} - i\delta_L \right)t} - 1 \right],$$ (16)

where

$$A \equiv \frac{\sqrt{\Gamma_a \Delta}}{\frac{\Gamma_a + \Gamma_b - \Delta}{2} - i\delta_L},$$ (17)

with $\delta_L \equiv \omega_L - \omega_a$. The resonance condition, $\delta_L = 0$, is necessary to maximize $|\psi(t)|$.

Now, the relevant degree of freedom is the size of the pulse, $\Delta^{-1}$, given fixed $\Gamma_a$ and $\Gamma_b$. If $\Delta^{-1} \to \infty$, the pulse is very long and the excitation is arbitrarily small: $|\psi(t)|^2 \propto \Delta \to 0$, as discussed in option (2). If $\Delta^{-1} \to 0$, the pulse is arbitrarily short, also causing the excitation probability to vanish: $|\psi(t)|^2 \propto \Delta^{-1} \to 0$. This is the worst-case scenario, since both the size and the duration of the atomic excitation are negligible. For intermediate pulse sizes, $\Delta \sim \Gamma_a \sim \Gamma_b$, we find from Equation (16) that

$$|\psi(t)| \leq 2\sqrt{\frac{\Gamma_a}{\Gamma}} \left( \frac{\Delta}{\Gamma} \right)^{-\frac{1}{2} \frac{\Delta + \Gamma}{\Delta - \Gamma}},$$ (18)

where we have defined $\Gamma = \Gamma_a + \Gamma_b$. The excitation probability is, therefore, maximized at $\Delta = \Gamma$ when the duration of the pulse is equal to the atomic lifetime (with the cost that its duration in time is strongly reduced with respect to the monochromatic regime, where $\Delta^{-1} \to \infty$). This is the scenario of option (1).

By substituting Equation (16) into (13), we find that

$$p_{a \to b}(\infty) = \frac{4\Gamma_a \Gamma_b}{\Gamma(\Gamma + \Delta)}.$$ (19)

This clearly shows that option (2) is the correct one. That is, the transition probability is maximized when the duration of the pulse is maximal ($\Delta/\Gamma \to 0$), even though the excitation probability is vanishingly small in that same situation.

The fact that $p_{a \to b}(\infty) \to 1$ if $\Gamma_a = \Gamma_b$ and $\Delta \to 0$ (i.e., in the monochromatic limit, which corresponds to a very large duration of the pulse, $\Delta^{-1} \to \infty$), as shown by Equation (19), is a signature of the beneficial effect of quantum coherence in this process. To make that more clear, we compare the "absorption-plus-emission" and the "quantum-

coherent" pictures. In the absorption-plus-emission picture, the photon excites the atom, which spontaneously decays towards either $|a\rangle$ or $|b\rangle$. If $\Gamma_a = \Gamma_b$, we expect that the initially excited atom has equal probabilities (1/2) of being found at $|a\rangle$ or $|b\rangle$ for $t \to \infty$. Indeed, this is what we get from our model if we set the initial state to $|e, 0\rangle$. We also get that (asymptotic probabilities of 50%) if we set $|a, 1_a\rangle$ as the initial state and choose $\Delta = \Gamma = 2\Gamma_a$, which corresponds to a relatively high excitation probability (of $|\psi(t)|^2 \leq 2e^{-2} \approx 0.27$; note that we could, in principle, find other pulse shapes that would lead to higher atomic excitations, with the inverted exponential being the best example [23], thus making spontaneous emission effects even more pronounced). However, the absorption-plus-emission picture does not capture the physics of the $\Delta \to 0$ limit; because the single-photon pulse is normalized, $\mathcal{N} = \sqrt{2\pi\varrho\Delta}$, a vanishing $\Delta$ also implies a vanishing field strength $\mathcal{N}$, so the excitation probability is negligibly small, $|\psi(t)|^2 \ll 1$, explaining why the effect of spontaneous emission is also negligible in this regime. Instead, the dynamics of the atom and the field (for $0 < t < \infty$) can be approximately described by the entangled (quantum-coherent) state $\phi_a(t)|a, 1_a\rangle + \phi_b(t)|b, 1_b\rangle$, with $\phi_a(0) = 1$ and $\phi_b(\infty) \to 1$. Along with this dynamic picture of the process, the thermodynamic picture (discussed below) will also be useful in clarifying why the $\Delta \to 0$ regime is so peculiar.

We are now left with the following question: If the excitation is negligible, is the work transferred from the photon to the atom also negligible, therefore violating the classical dissipative adaptation hypothesis?

To answer that question, we have to define the work performed by the single-photon drive. Classically, the work of a time-varying classical electric field $E(t)$ acting on a classical dipole $D(t) = qx(t)$ is given by $W = \int F\dot{x}\, dt = \int qE\dot{x}\, dt = \int \dot{D}E\, dt$. Here, we define the average work performed by a single-photon pulse on a quantum dipole by employing the Heisenberg picture:

$$\langle W \rangle = \int_0^\infty \langle (\partial_t D(t)) E_{\text{in}}(t) \rangle\, dt, \tag{20}$$

where $E_{\text{in}}(t) = \sum_\omega i\epsilon_\omega a_\omega e^{-i\omega t} + \text{h.c.}$ is the incoming field. The field produced by the atom that acts back on the atom itself gives rise to heat dissipation in our model. The dipole operator is given by $D(t) = U^\dagger D U$, with $D = \sum D_{ea}\sigma_a + \text{h.c.}$, so that $\hbar g_a = D_{ea}\epsilon_{\omega_a}$.

Using integration by parts, we can rewrite Equation (20) as $\langle W \rangle = -\int_0^\infty \langle D(t)\partial_t E_{\text{in}}(t) \rangle dt$. Within the rotating-wave approximation, this gives us that

$$\begin{aligned}
\langle W \rangle = -\int_0^\infty dt\, (i\hbar) \sum_\omega (-i\omega)g_a \langle \sigma_a^\dagger(t)a_\omega \rangle e^{-i\omega t} \\
+ (-i\omega)g_b \langle \sigma_b^\dagger(t)b_\omega \rangle e^{-i\omega t} \\
+ \text{c.c.},
\end{aligned} \tag{21}$$

where c.c stands for the complex conjugate. Choosing $|1_a\rangle$ as the initial state of the field implies that $\langle \sigma_b^\dagger(t)b_\omega \rangle = 0$. The non-zero correlation function is

$$\langle \sigma_a^\dagger(t)a_\omega \rangle = \langle U^\dagger \sigma_a^\dagger U a_\omega | a, 1_a \rangle \tag{22}$$

$$= \langle U^\dagger \sigma_a^\dagger U \phi_\omega(0) | a, 0 \rangle \tag{23}$$

$$= \phi_\omega(0)\langle a, 1_a | U^\dagger | e, 0 \rangle \tag{24}$$

$$= \phi_\omega(0)\psi^*(t). \tag{25}$$

It is worth noting that, because the work performed by the photon on the atom depends on the correlation calculated above, it becomes clear that the more in phase the atom is with respect to its driving field, the more work it will absorb. But this atom–field synchronization requires the field to be as monochromatic as possible, explaining why the $\Delta \to 0$ limit is so special for both the dynamics and for the thermodynamics of the process. We thus get that

$$\langle W \rangle = -\hbar g_a \int_0^\infty dt\, \psi^*(t)i\partial_t \phi_a(-ct, 0) + \text{c.c.}. \tag{26}$$

For a pulse of central frequency $\omega_L$ and a general envelope shape, we define $\phi_a(-ct,0) = \phi_a^{env}(-ct,0)\exp(-i\omega_L t)$. The derivative of the fast-oscillating part gives rise to

$$\langle W \rangle = \hbar\omega_L \int_0^\infty dt \, (-2g_a \text{Re}[\psi^*(t)\phi_a(-ct,0)]). \tag{27}$$

The derivative of the slowly varying part is related to $\text{Im}[\psi^*(t)\exp(-i\omega_L t)]$, since $\partial_t \phi_a^{env}(t)$ is real. From Equation (15), it follows that $\text{Im}[\psi^*(t)\exp(-i\omega_L t)] = 0$ if $\delta_L = 0$. This shows that, at resonance, only the absorptive contribution remains, while the dispersive (reactive) contribution vanishes.

From Equation (14), we can derive the dynamics of the excited-state population $|\psi(t)|^2$,

$$\partial_t |\psi(t)|^2 = -\Gamma|\psi(t)|^2 - 2g_a \text{Re}[\psi^*(t)\phi_a(-ct,0)]. \tag{28}$$

Substituting this back into Equation (27) gives us that

$$\langle W \rangle = \hbar\omega_L \int_0^\infty \Gamma|\psi(t)|^2 dt, \tag{29}$$

where we used $|\psi(0)|^2 = |\psi(\infty)|^2 = 0$. By comparing Equations (29) and (13), we immediately see that

$$p_{a\to b}(\infty) = \frac{\Gamma_b}{\Gamma}\frac{\langle W \rangle}{\hbar\omega_L}, \tag{30}$$

which is the quantum dissipative adaptation relation for a generic single-photon pulse (not restricted to the exponential profile). This result answers the question that we raised above by showing that, even though the excitation probability is negligible in the highly monochromatic case, the transferred work is maximized in that regime along with the transition probability from $|a\rangle$ to $|b\rangle$.

It is important to clarify that the purpose of this section is mainly to provide completeness for our text, as a significant part of it has also been discussed in [13]. Still, Equation (18) represents an original result. Also, here, we established the contrast between the "absorption-plus-emission picture" and the "quantum-coherent picture", which will help us below to make sense of how the average work behaves in the presence of multiple photons. Finally, the explicit calculation of the work departing from the Heisenberg picture in the context of the present model also goes beyond [13]. Our motivation for choosing the Heisenberg formalism here (rather then the Schrödinger picture from [13]) is to facilitate the thermodynamic analysis in the presence of multiple pulses and especially to evidence the additive property of the work, which is a crucial step in our main results. Because of the Heisenberg formalism, in Equation (30), we see the dependence of $\omega_L$, whereas in [13], $\omega_a$ appears instead. This will be relevant if we want to depart from the resonance condition $\omega_L = \omega_a$ that is assumed here.

### 3.2. Quantum Dissipative Adaptation for Cascaded Photons

By cascaded interactions, we mean that consecutive non-overlapping and uncorrelated single-photon pulses are driving the atom. Under this assumption, we have that either the first photon promotes the transition or it leaves the atom unaltered and the second photon enables the transition, and so on. Asymptotically ($t \to \infty$), the total probability transition $p_{a\to b}^{(T,N)}$ after $N$ pulses have driven the atom is given by

$$p_{a\to b}^{(T,N)} = \sum_{k=1}^{N} \prod_{j=1}^{k-1} p_{a\to a}^{(j)} p_{a\to b}^{(k)}. \tag{31}$$

Here, $p_{a\to a}^{(j)} = 1 - p_{a\to b}^{(j)}$ is the probability that the atom remains in state $|a\rangle$ during the interaction with the $j$-th pulse, and $p_{a\to b}^{(k)}$ is the transition probability due to the $k$-th photon. The degrees of freedom considered in this model are the linewidths $\{\Delta_k\}$ of the $N$ photon pulses.

The average work in this cascaded process is additive:

$$\langle W^{(N)} \rangle = \sum_{k=1}^{N} \langle W \rangle_k, \tag{32}$$

where $\langle W \rangle_k$ is the average work performed by the $k$-th photon of linewidth $\Delta_k$. The average work performed by the $k$-th photon is given by

$$\langle W \rangle_k = p_a^{(k-1)} \langle W \rangle_a^{(k)} + p_b^{(k-1)} \langle W \rangle_b^{(k)}, \tag{33}$$

which depends on the probability $p_i^{(k-1)}$ that the previous photon has left the atom in state $|i\rangle$. Also, we define $\langle W \rangle_a^{(k)}$ to evidence that the initial state of the system for which the work is being computed is $|a, 1_a\rangle$, and this is $|b, 1_a\rangle$ for $\langle W \rangle_b^{(k)}$. That is, all of the photons are initially equally polarized; only the atom may be in a different state. Thus, we find that $\langle W \rangle_b^{(k)} = 0$ for all $k$, implying that

$$\langle W^{(N)} \rangle = \sum_{k=1}^{N} p_a^{(k-1)} \langle W \rangle_a^{(k)}. \tag{34}$$

Because there is no physical mechanism in our model that makes the atom jump back from $|b\rangle$ to $|a\rangle$ (due to the zero temperature), the only possible path for it to be found at $|a\rangle$ is that where the atom has never left state $|a\rangle$ throughout its entire history. This means that $p_a^{(k-1)}$ is a product of the probabilities that all of the previous photons have also left the atom at $|a\rangle$ so that

$$\langle W^{(N)} \rangle = \sum_{k=1}^{N} \left( \prod_{j=1}^{k-1} p_{a \to a}^{(j)} \right) \langle W \rangle_a^{(k)}. \tag{35}$$

Using the quantum dissipative adaptation relation for each single photon, namely, $\langle W \rangle_a^{(k)} = p_{a \to b}^{(k)} \hbar \omega_L \Gamma / \Gamma_b$, we have that

$$\langle W^{(N)} \rangle = \left( \sum_{k=1}^{N} \prod_{j=1}^{k-1} p_{a \to a}^{(j)} p_{a \to b}^{(k)} \right) \hbar \omega_L \Gamma / \Gamma_b. \tag{36}$$

By employing Equation (31), we finally find that

$$p_{a \to b}^{(T,N)} = \frac{\Gamma_b}{\Gamma} \frac{\langle W^{(N)} \rangle}{\hbar \omega_L}, \tag{37}$$

showing that the quantum dissipative adaptation remains valid for $N$ cascaded photons of arbitrary shapes.

### 3.3. Long Versus Short Pulses

Is it true that two pulses always increase the degree of organization of the atom? Or can we find a single photon long enough to produce an equivalent result? With these questions in mind, we take a closer look at the case of $N = 2$.

To answer that question, we analyze

$$\begin{aligned} p_{a \to b}^{(T,2)} &= p_{a \to a}^{(1)} p_{a \to b}^{(2)} + p_{a \to b}^{(1)}, \\ &= (1 - p_{a \to b}^{(1)}) p_{a \to b}^{(2)} + p_{a \to b}^{(1)}. \end{aligned} \tag{38}$$

We assume from here on that our photons have exponential envelope profiles, as described by Equation (9), with two generally distinct linewidths $\Delta_1$ and $\Delta_2$. Thus, Equation (19) is valid, and we have that

$$p_{a \to b}^{(k)} = \frac{r}{1 + \Delta_k / \Gamma}, \tag{39}$$

where $r = 4 \Gamma_a \Gamma_b / \Gamma^2$, and $\Gamma = \Gamma_a + \Gamma_b$. In order to reiterate the insight provided by the notion of a quantum dissipative adaptation, namely, that the dynamics of non-equilibrium

self-organization (transition probabilities) are intimately related to the thermodynamics (average work cost), we also explicitly calculate the average work, $\langle W^{(2)} \rangle / \hbar \omega_L = (\Gamma / \Gamma_b) p_{a \to b}^{(T,2)}$, for each case that we analyze below.

Let us take the limit of two pulses that are both very short in time and space (i.e., highly broadband), $\Delta_1 \gg \Gamma$ and $\Delta_2 \gg \Gamma$. In that case,

$$p_{a \to b}^{(k)} \approx r \Gamma \tau_k \ll 1, \tag{40}$$

where $\tau_k = \Delta_k^{-1}$ is the typical time duration of the pulse. The total probability now reads

$$p_{a \to b}^{(T,2)} \approx r \Gamma (\tau_1 + \tau_2) = r \Gamma \tau_{\text{eff}} \approx p_{a \to b}^{(1)}|_{\tau_{\text{eff}}}, \tag{41}$$

where we have neglected the second-order term $\mathcal{O}[\Gamma^2 \tau_1 \tau_2]$. This shows that two short pulses are indeed equivalent to a single longer pulse with an effective time duration given by the sum of the individual pulses, $\tau_{\text{eff}} = \tau_1 + \tau_2$, as we could intuitively expect. In that case, the work is given by $\langle W^{(2)} \rangle / \hbar \omega_L \approx \langle W^{(1)} \rangle_{\text{eff}} / \hbar \omega_L = 4 \Gamma_a (\tau_1 + \tau_2)$, which only depends on $\Gamma_a$. Because $\Gamma_a$ describes the absorption channel, this result is typical of the "absorption-plus-emission" picture that we discussed in Section 3.1.

We now consider the opposite limit, namely, two very long pulses in time (highly monochromatic), $\Delta_1 \ll \Gamma$ and $\Delta_2 \ll \Gamma$. We have that

$$p_{a \to b}^{(k)} \approx r, \tag{42}$$

which does not depend on $\Delta_k$. Hence,

$$p_{a \to b}^{(T,2)} \approx (1 - r) r + r, \tag{43}$$

where the parameter $r$ is bounded to $r = 4 \Gamma_a \Gamma_b / (\Gamma_a + \Gamma_b)^2 \leq 1$.

Depending on $r$, we find two possible scenarios. If the two decay rates are identical ($\Gamma_a = \Gamma_b$, thus $r = 1$), we have that $p_{a \to b}^{(T,2)} = r = p_{a \to b}^{(T,1)}$. This shows that two pulses can be replaced by a single one again, as in the case of two short pulses. The work can now be expressed as $\langle W^{(2)} \rangle / \hbar \omega_L \approx \langle W^{(1)} \rangle / \hbar \omega_L \approx 4 \Gamma_a / (\Gamma_a + \Gamma_b)$, which depends on both $\Gamma_a$ and $\Gamma_b$, since the "quantum-coherent" regime involves a superposition between these two channels (see Section 3.1). However, in the far more typical case where $\Gamma_a \neq \Gamma_b$, we have that $r < 1$, so

$$p_{a \to b}^{(T,2)} > r = p_{a \to b}^{(T,1)}. \tag{44}$$

We see that a single long pulse saturates the transition probability to a value below unity ($p_{a \to b}^{(1)} = r < 1$), and the addition of a second pulse is the only way to improve driven self-organization. Finally, the work in this case is $\langle W^{(2)} \rangle / \hbar \omega_L \approx 8 \Gamma_a / \Gamma - 16 \Gamma_a^2 \Gamma_b / \Gamma^3 > \langle W^{(1)} \rangle / \hbar \omega_L \approx 4 \Gamma_a / \Gamma$. In this case, $\langle W^{(2)} \rangle$ depends on higher orders of $\Gamma_a$ and $\Gamma_b$.

For $0 < r < 1$ and considering $N > 2$ photons of arbitrary linewidths, we have that

$$\begin{aligned} p_{a \to b}^{(T,N)} &= \frac{r}{1 + \Delta_1 / \Gamma} + \sum_{k=2}^{N} \prod_{j=1}^{k-1} \left( 1 - \frac{r}{1 + \Delta_j / \Gamma} \right) \frac{r}{1 + \Delta_k / \Gamma} \\ &> \frac{r}{1 + \Delta_1 / \Gamma}, \end{aligned} \tag{45}$$

which means that the degree of self-organization only increases with $N$.

Note that $p_{a \to b}^{(T,N)} \leq 1$ for $N \to \infty$, as we can see by taking the extreme limit where all of the photons are highly monochromatic, thus maximally increasing the transition probability. In that case, $\Delta_k / \Gamma \to 0$ for all $k$, so

$$p_{a \to b}^{(T,N)} \approx r + r \sum_{k=1}^{N-1} (1-r)^k$$
$$= r + r\left( (1-r)\, \frac{1-(1-r)^{N-1}}{r} \right)$$
$$= 1 - (1-r)^N$$
$$\leq 1. \tag{46}$$

We used the notion that, in the geometric series, $\sum_{n=1}^{M} a_1 q^{n-1} = a_1(1-q^M)/(1-q)$, with $a_1 = q = 1 - r$ and $M = N - 1$. So, the series in our quantum model of $N$ cascaded photons converges as it should.

## 4. Discussion

We have investigated the validity of the dissipative adaptation in a quantum model where each $\Lambda$ atom in an ensemble is driven by a cascaded sequence of $N$ single-photon pulses. We have found that, the more energy the atoms are given, the more organized they become, which is reminiscent of the evolutionary dynamics of living systems. Our model generalizes the previous ones where the quantum dissipative adaptation was found, namely, its original proposal in [13], as well as its applications in the self-replication of quantum artificial organisms [14] and the emergence of energy-seeking and energy avoidance in classically driven dissipative few-level quantum systems [16]. We have also found that, in the typical case where the atom has unequal decay rates, the level of organization achieved with two pulses matches that with a single one, provided that the two photons are short enough; otherwise, the presence of multiple photons becomes a resource for raising the degree of self-organization.

The cascaded-photons hypothesis used here allowed us to simplify our model, giving us the statistical perspective that we employed instead of solving the far more complicated quantum dynamics in the $N$-excitation subspace. Part of our results could have been anticipated by thinking in terms of Landauer's principle (originally classical, but also verified for quantum systems, even at zero temperatures [24]). Landauer's principle imposes a minimal energy cost (in terms of heat dissipation) for a memory erasure to be performed (entropy reduction or a transition from a mixed to a pure state of the atomic system, to translate it to the the present case). As an advantage, our model provides an exact value for the energetic consumption, thus being more precise than Landauer's bound. To compute work, even in our simplified model, we made a non-trivial assumption that perhaps should be discussed more explicitly. Namely, the assumption that each new photon pulse meets an atom in a mixed state (as assumed in Equation (33)) is necessary for us to compute work here, but this could be revisited in a more general model. This is because the linearity of quantum mechanics implies that the asymptotic state after a single-photon scattering is generally given by an entangled state (as discussed in Section 3.1). Only after tracing out the field degrees of freedom did we obtain the mixed state for the atom that we assumed. Understanding how the work behaves in the context of more general quantum matter–field states, including all of the quantum correlations that naturally arise in quantum models, is left as an open question.

As a perspective, we plan to study the case where multiple pulses overlap. This could lead to irreversible stimulated emissions, as shown in [20], eventually forcing the atom to go back to its initial state $|a\rangle$, thereby reducing the chances of self-organization happening. Understanding not only whether this is the case but also how to compute work in that more complex scenario, as well as further testing the quantum dissipative adaptation hypothesis, is a challenging and timely question. In the field of quantum thermodynamics, fundamental questions concerning the definition of work are currently being debated [25–30], as well as its applications in the energetic characterization of quantum information processing [26–28]. While Ref. [25] proposed a Heisenberg formalism for defining mechanical work as an operator in quantum mechanics, it has been applied only

to isolated systems so far. The fact that we are considering a driving force (coming from the single-photon pulse) in addition to a dissipative force (leading to spontaneous emission and coming from the continuum of frequencies in the electromagnetic environment) represents a step further in the thermodynamic description of autonomous quantum systems (i.e., those where no explicit time-dependent Hamiltonian is assumed from the outset). The energetic characterization of quantum information processing of autonomous systems is a flourishing field of research, and our Heisenberg approach comes out as a promising method to that end. Moreover, the fact that we are considering Λ three-level atoms (as is also key in [31], instead of the two-level atoms studied in [26–28]) also deserves further investigation as far as the energetics of quantum information processing are concerned.

In the long run, the insight provided not only by the dissipative adaptation perspective but also by more general biologically inspired non-equilibrium self-organization could open radically new paths for classical and quantum information processing, which could potentially be useful for their resilience, self-restoring capacities, and exquisite thermodynamic efficiencies. In the classical case, we see promising discussions of alternative computational schemes with stochastic and biologically inspired systems in [32–35], for instance. In the quantum case, we would like to go beyond our discussions here and in [14,16] (concerned, respectively, with quantum cloning and heat management) so as to highlight the remarkable biologically inspired model for quantum error correction in [36] while also employing non-equilibrium light–matter interactions.

**Author Contributions:** T.G., T.W. and D.V. contributed equally to this work. All authors have read and agreed to the published version of the manuscript.

**Funding:** This work was funded by CNPq INCT-IQ 465469/2014-0 and by CNPq 402074/2023-8. T.G. was funded by CAPES.

**Institutional Review Board Statement:** Not applicable.

**Informed Consent Statement:** Not applicable.

**Data Availability Statement:** Data are contained within the article.

**Conflicts of Interest:** The authors declare no conflicts of interest. The funders had no role in the design of the study, in the collection, analyses, or interpretation of data, in the writing of the manuscript, or in the decision to publish the results.

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
