# Peer review of "Quantum Dissipative Adaptation with Cascaded Photons"

_photonics, doi:10.3390/photonics11010041_

Round 1

Reviewer 1 Report

Comments and Suggestions for Authors

The authors analyze a dissipation-assisted transition across an energy barrier (modeled by a lambda atom with polarization-selective transitions between the excited state and the two ground states interacting with a single photon pulses) and analyze the relation between the work deposited by the photons and the transition rates, in the context of dissipative adaptation.

They show that an earlier result expressing the nonequilibrium transition rate induced by a single photon pulse as proportional to the work input performed by the photon on the atom can be generalized for the case of N independent photons, and that there is a real advantage in having more than one single photon pulse in term of achievable net transition rate.

The results are interesting and provide some new insights into nonequilibrium mechanisms. The article is well written. I recommend publication after minor corrections that I suggest belowM

- Typo l42: then → than

- The last sentence of the first paragraph of the introduction sounds really obscure to me. What does it mean for transition probabilities to be “most relevant”? How a work cost can dominate a probability? I would recommend that the author rewrite this sentence with maybe more transparent formulations of the connection between work cost and nonequilibrium transitition probabilities that is postulated by the dissipative adaptation hypothesis.

- Around Eq. (17), (18): Can the author detail a bit more why option 1 leads to a smaller transition probability? This conclusion is not clear to me from the sole comparison between Eq.(17) and Eq.(18).

Author Response

The authors analyze a dissipation-assisted transition across an energy barrier (modeled by a lambda atom with polarization-selective transitions between the excited state and the two ground states interacting with a single photon pulses) and analyze the relation between the work deposited by the photons and the transition rates, in the context of dissipative adaptation.

They show that an earlier result expressing the nonequilibrium transition rate induced by a single photon pulse as proportional to the work input performed by the photon on the atom can be generalized for the case of N independent photons, and that there is a real advantage in having more than one single photon pulse in term of achievable net transition rate.

The results are interesting and provide some new insights into nonequilibrium mechanisms. The article is well written. I recommend publication after minor corrections that I suggest below

We thank the Referee for their kind consideration.

- Typo l42: then → than

Indeed. That has been corrected. We thank the Referee.

- The last sentence of the first paragraph of the introduction sounds really obscure to me. What does it mean for transition probabilities to be “most relevant”? How a work cost can dominate a probability? I would recommend that the author rewrite this sentence with maybe more transparent formulations of the connection between work cost and nonequilibrium transition probabilities that is postulated by the dissipative adaptation hypothesis.

We have added a new paragraph to the introduction trying to make these ideas more clear. We agree that the original version was not clear enough.

In essence, the idea is to compare (nonequilibrium) dissipative adaptation with the (equilibrium) Boltzmann distribution.

If in Boltzmann distribution the key quantity is the probability of a given state, p_k, in the dissipative adaptation, it is the transition probability between a pair of states, p_i->j.

Also, if in the Boltzmann distribution the energy of the state E_k governs the tendencies, p_k ~ exp(- E_k / kB T), in the dissipative adaptation the work W plays that role too, p_i->j ~ <exp(W / kB T)>_i->j.

- Around Eq. (17), (18): Can the author detail a bit more why option 1 leads to a smaller transition probability? This conclusion is not clear to me from the sole comparison between Eq.(17) and Eq.(18).

We have added a new paragraph after eq.(17) [of the original version, corresponding to eq.(19) of the revised version] to better explain that point.

Mathematically, the main point is that when we achieve the highest possible values of |\psi(t)|, this comes at the cost of a very thin peak in time, hence the area (integral in time) is reduced.

Physically, this can be understood by comparing the “absorption-plus-emission” picture with the “quantum-coherent” picture, as we describe in our revised version of the manuscript.

We hope that the Referee now finds the present version of our manuscript suitable for publication.

Yours sincerely,

The authors

Reviewer 2 Report

Comments and Suggestions for Authors

The authors study an atomic Lambda-system formed by two almost degenerate levels (“ground states”, “a” and “b”) coupled to an excited states (“e”).

From the excited state “e”, the system can relax by emitting a photon whose polarization depends on the final state, i.e. “a” and “b”.

They discuss the case when an incoming single-photon pulse, with given polarisation and at frequency omega_a, is resonant to the energy difference between “e” and “a” and they calculate the probability of emitting a  single-photon pulse at frequency omega_b, resonant to the energy difference between “e” and “b”, namely the photon at omega_a has been absorbed and another photon has been emitted a frequency \omega_b, with the concomitant switching of the atom from the state “a” to the state “b”.

The motivation of this model is for testing the self-organization dynamics: if we consider an ensemble of mixed atoms in the two possible relaxed states, “a” and “b”, after many pulses one expects to find all the atoms in the state “b”.

The fundamental quantity is the conditional probability that, starting from “a” state and for the impinging photons at omega_a, the system 

switches to the state “b”.

Using the Hamiltonian in the rotation wave approximation, the dynamics can be solved exactly in analytic way owing to the conservation 

of the quanta of excitation in the system.

They also relate the probability to the work done by the field on the atomic dipole.

The work is interesting but  I have some questions and recommendations.

1) 

It seems to me that the results, given in the section 3.1 have been already presented in Ref.8, 

given the tittle of the section “Revisiting the single-photon wavepacket scenario”.

It is not clear what are the original results in this part.

Maybe the authors can write explicitly what is the new formulas and results, as compared to Ref.8

2) 

If I take the formula Eq. 18, assuming that the dissipation rates  are the same Gamma_a=Gamma_b, and the energy difference between the two levels “a” and “b” is zero, by reducing the duration of the pulse,

then the probability is one, namely the photon at omega_a is absorbed at 100% and it is re-emitted as a photon at omega_b.

This is counterintuitive as the photon could be re-emitted at frequency \omega_a.

Can the authors give a physical argument about this result?

In the most symmetric case, I would intuitively expect half and half to be the system in state “a” or “b”.

3) in the section 3.2, the authors discuss the probability of switching when N single-photon pulse.

non-overlapping and uncorrelated.

Having the individual probability p_ab, this problem reduces to a classical  statistical one based on the combinatorial analysis.

I invite the authors to stress what are the new results and what could be directly inferred from the previous results reported in the literature.

4) 

in the section 3.3, the author aim to demonstrate whether a single pulce can reproduce the same result of two pulses.

They give a positive answer but it is not clear the connection with the main context, namely how this result 

is related to the idea Quantum dissipative adaptation.

Maybe the authors can shortly comment their results in this perspective to help the reader to appreciate the results.

Author Response

The authors study an atomic Lambda-system formed by two almost degenerate levels (“ground states”, “a” and “b”) coupled to an excited states (“e”).

From the excited state “e”, the system can relax by emitting a photon whose polarization depends on the final state, i.e. “a” and “b”.

They discuss the case when an incoming single-photon pulse, with given polarization and at frequency omega_a, is resonant to the energy difference between “e” and “a” and they calculate the probability of emitting a  single-photon pulse at frequency omega_b, resonant to the energy difference between “e” and “b”, namely the photon at omega_a has been absorbed and another photon has been emitted a frequency \omega_b, with the concomitant switching of the atom from the state “a” to the state “b”.

The motivation of this model is for testing the self-organization dynamics: if we consider an ensemble of mixed atoms in the two possible relaxed states, “a” and “b”, after many pulses one expects to find all the atoms in the state “b”.

The fundamental quantity is the conditional probability that, starting from “a” state and for the impinging photons at omega_a, the system

switches to the state “b”.

Using the Hamiltonian in the rotation wave approximation, the dynamics can be solved exactly in analytic way owing to the conservation

of the quanta of excitation in the system.

They also relate the probability to the work done by the field on the atomic dipole.

We thank the Referee for their thoughtful reading of our manuscript.

The work is interesting but I have some questions and recommendations.

1)

It seems to me that the results, given in the section 3.1 have been already presented in Ref.8,

given the tittle of the section “Revisiting the single-photon wavepacket scenario”.

It is not clear what are the original results in this part.

Maybe the authors can write explicitly what is the new formulas and results, as compared to Ref.8

We thank the Referee for pointing that out. We have added a paragraph at the end of sec.3.1 to explicitly state what is new.

2)

If I take the formula Eq. 18, assuming that the dissipation rates  are the same Gamma_a=Gamma_b, and the energy difference between the two levels “a” and “b” is zero, by reducing the duration of the pulse,

then the probability is one, namely the photon at omega_a is absorbed at 100% and it is re-emitted as a photon at omega_b.

This is counterintuitive as the photon could be re-emitted at frequency \omega_a.

Can the authors give a physical argument about this result?

In the most symmetric case, I would intuitively expect half and half to be the system in state “a” or “b”.

This is a very important point, in fact. We have now added a new paragraph to answer this question.

The key thing is that, depending on the properties of the incoming pulse, two distinct pictures emerge.

If absorption is pronounced (that is, |\psi(t)|^2 achieves non-negligible values), then spontaneous emission will be significant. If spontaneous emission is the main mechanism, the “half-half” intuition is correct.

However, if absorption is negligible (that is, |\psi(t)|^2 << 1 for all times), then spontaneous emission is not the main effect. This can happen either for a very monochromatic pulse (\Delta << \Gamma) or for a very broad pulse (\Delta >> \Gamma). In the former, a quantum-coherent effect will allow for a transition probability closer to 100%; in the latter, almost no dynamics takes place at all, so the unbalance goes the other way around, namely, the transition probability is closer to 0 %.

3) in the section 3.2, the authors discuss the probability of switching when N single-photon pulse.

non-overlapping and uncorrelated.

Having the individual probability p_ab, this problem reduces to a classical  statistical one based on the combinatorial analysis.

I invite the authors to stress what are the new results and what could be directly inferred from the previous results reported in the literature.

We have considerably extended the discussion section, so as to clarify the significance of our results with respect to the literature.

4)

in the section 3.3, the author aim to demonstrate whether a single pulse can reproduce the same result of two pulses.

They give a positive answer but it is not clear the connection with the main context, namely how this result

is related to the idea Quantum dissipative adaptation.

Maybe the authors can shortly comment their results in this perspective to help the reader to appreciate the results.

We thank the referee for pointing that out. We have now added in sec. 3.3 the calculations and analysis of the work transfer from the photon to the atom, so as to provide a better connection with the main context.

We hope that the Referee now finds the present version of our manuscript suitable for publication. 

Yours sincerely,

The authors